# Preparation of photonic molecular trains via soft-crystal polymerization of lanthanide complexes

Pedro Paulo Ferreira da Rosa [1], Yuichi Kitagawa[2,3], Sunao Shoji [2,3], Hironaga Oyama[4], Keisuke Imaeda[5], Naofumi Nakayama [6], Koji Fushimi [2], Hidehiro Uekusa[4], Kosei Ueno [5], Hitoshi Goto [7] & Yasuchika Hasegawa [2,3]✉

Soft-crystals are defined as flexible molecular solids with highly ordered structures and have attracted attention in molecular sensing materials based on external triggers and environments. Here, we show the soft-crystal copolymerization of green-luminescent Tb(III) and yellow-luminescent Dy(III) coordination centers. Soft-crystal polymerization is achieved via transformation of monomeric dinuclear complexes and polymeric structures with respect to coordination number and geometry. The structural transformation is characterized using single-crystal and powder X-ray diffraction. The connected Tb(III) crystal-Dy(III) crystal show photon energy transfer from the Dy(III) centre to the Tb(III) centre under blue light excitation (selective Dy(III) centre excitation: $460 \pm 10$ nm). The activation energy of the energy transfer is estimated using the temperature-dependent emission lifetimes and emission quantum yields, and time-dependent density functional theory (B3LYP) calculations. Luminescence-conductive polymers, photonic molecular trains, are successfully prepared via soft-crystal polymerization on crystal media with remarkable long-range energy migration.

[1] Graduate School of Chemical Sciences and Engineering, Hokkaido University, Sapporo, Hokkaido 060-8628, Japan. [2] Faculty of Engineering, Hokkaido University, Sapporo, Hokkaido 060-8628, Japan. [3] Institute for Chemical Reaction Design and Discovery (WPI-ICReDD), Hokkaido University, Sapporo, Hokkaido 001-0021, Japan. [4] School of Science, Tokyo Institute of Technology, Tokyo 152-8551, Japan. [5] Faculty of Science, Hokkaido University, Sapporo, Hokkaido 060-0810, Japan. [6] CONFLEX Corporation, Tokyo 108-0074, Japan. [7] Information and Media Center, Toyohashi University of Technology, Toyohashi, Aichi 441-8580, Japan. ✉email: hasegaway@eng.hokudai.ac.jp

Flexible molecular solids with highly ordered structures are denoted soft crystals[1]. Photonic soft crystals recently attracted significant attention in the fields of vapochromism[2–5], mechanochromism[6–9], and elastic single-crystal chemistry[10–12]. Remarkable photonic soft crystals, such as the transformative luminescent Pt(II) complex syn-[Pt$_2$(bpy)$_2$(pyt)$_2$][PF$_6$]$_2$ (bpy = 2,2'-bipyridine, pyt = pyridine-2-thiolate) in the presence of ethanol vapor[13,14], a mechanochromic luminescent isocyanide Au(I) complex[6], and optical waveguides using elastic hexachlorobenzene crystals[15,16], have been reported. Historically, photochromic soft crystals were also studied[17,18]. Highly ordered materials with soft structures emerged as a new area of science and crystal engineering for the development of sophisticated responsive materials.

The structural transformations in soft crystals are induced by changes in molecular ordering under external stimuli via (1) phase transition, (2) molecular structural change, and (3) addition of external molecules[19]. These characteristic transformations are based on one single crystal. Inspired by the generation of novel properties during molecular assembly, we explored the merging of different crystals at the molecular level to form connected crystals. Here, we demonstrate the soft-crystal connection of different crystals using a transformative lanthanide coordination center to form novel photofunctional crystals. Generally, lanthanide coordination compounds, which are promising luminescent materials, exhibit photophysical properties derived from 4f–4f transitions[20–25]. The connected crystals may result in new functional materials, such as highly ordered nanocrystal semiconductors or crystal block copolymers.

The lanthanide coordination centers consist of central lanthanide ions and organic ligands. The lanthanide ions exhibit similar large ionic radii, resulting in the formation of similar structures regardless of the identity of the lanthanide ion[21,22,26,27]. These characteristics provide the necessary conditions for the connection of two crystals based on different lanthanide coordination centers via the same structural transformations. Currently, the predominant lanthanide complexes exhibit seven-, eight- or nine-coordinate structures[28]. We reported the formation of seven-coordinate [Eu(tmh)$_3$(py)$_1$] (tmh: 2,2,6,6-tetramethyl-3,5-heptanedionato; py: pyridine) and eight-coordinate [Eu(tmh)$_3$(py)$_2$][29], demonstrating the potentially transformative geometrical coordination structures of crystalline lanthanide complexes. Based on our structural findings, a seven-coordinate dinuclear complex [Ln$_2$(tmh)$_6$(4,4'-bpy)] (4,4'-bpy: 4,4'-bipyridine) with a simple bidentate 4,4'-bpy bridging ligand was selected for soft-crystal connection using a monodentate py trigger. Here, we use bulky photosensitized tmh and free-rotating 4,4'-bpy ligands in the formation of several metastable structures. Tb(III) ions bearing tmh ligands have been reported due to their bright emission and sensing properties[30–35]. Dy(III) ions also exhibit unique yellow emission and compatible excited energy levels that may match those of the Tb(III) ions[36]. Herein, we prepare an eight-coordinate coordination polymer [Ln(tmh)$_3$(4,4'-bpy)]$_n$ via transformation of a dinuclear complex [Ln$_2$(tmh)$_6$(4,4'-bpy)] under py vaporization. We also attempt to form Tb(III) and Dy(III) soft coordination crystals, denoted molecular trains (Fig. 1) to yield a photonic block crystal based on crystal-to-crystal long-range energy migration. Structural changes in the crystal are monitored using single-crystal and powder X-ray diffraction (PXRD), electrospray ionization mass spectrometry (ESI-MS), and elemental analysis. The photophysical properties are characterized using emission spectroscopy, temperature-dependent emission lifetimes, and quantum yields. Quantum chemical calculations are performed to study the excited states. The crystal-to-crystal energy migration is measured using a spatially resolved microspectroscopic system for emission detections at specific regions of the crystal.

In this work, soft-crystal polymerization between Tb(III) and Dy(III) coordination centers on the crystal surfaces is achieved. The dynamic polymerization of the lanthanide complexes in crystal media results in the merging of the Tb(III) and Dy(III) complexes due to their identical structures and transformations. After the connected crystals are formed via py vaporization, the system shows long-range energy transfer from the Dy(III) crystal to the Tb(III) crystal (~150 μm).

## Results

**Synthesis and structures.** Lanthanide complexes containing bulky tmh and phosphine oxide ligands reportedly exhibit seven-coordinate structures[28,30–33]. These structures are thermodynamically stable and do not transform into eight-coordinate coordination polymers. To promote polymerization in crystal media, we synthesized a metastable lanthanide complex using free-rotating 4,4'-bpy. Dinuclear [Ln$_2$(tmh)$_6$(4,4'-bpy)] (Ln: Tb(III) and Dy(III)) complexes (Ln-dinuclear) and coordination polymers [Ln(tmh)$_3$(4,4'-bpy)]$_n$ (Ln-polymer) were independently synthesized using high and low molar ratios of the precursor [Ln$_2$(tmh)$_6$] (Supplementary Methods and Supplementary Fig. 1) and 4,4'-bpy, respectively. Different molar ratios of the reactants yield different products[37]. Dinuclear complexes are synthesized via the complexation of 4,4'-bpy with [Ln$_2$(tmh)$_6$] using excess [Ln$_2$(tmh)$_6$] (1.5 eq.) in methanol (Supplementary Fig. 2). Conversely, coordination polymers are synthesized via complexation of 4,4'-bpy with [Ln$_2$(tmh)$_6$] using excess 4,4'-bpy (2 eq.) in methanol (Supplementary Fig. 2). Both solutions yield transparent single crystals.

To investigate the structure of each crystal, single-crystal X-ray diffraction was performed (Supplementary Note 1 and Supplementary Tables 1, 2). Ln-dinuclear consists of two lanthanide cores that each bear three tmh ligands and are connected by one 4,4'-bpy bridging ligand. Both lanthanide cores exhibit seven-coordinate structures (Fig. 2a). Ln-dinuclear displays polymorphism, with two similar seven-coordinate dinuclear structures (Supplementary Note 2, Supplementary Fig. 3, and Supplementary Table 2). For Tb-polymer, the Tb(tmh)$_3$ center is coordinated to two 4,4'-bpy bridging ligands (Fig. 2b), resulting in the formation of a coordination polymer due to the increase in coordination number.

The pseudo-coordination polyhedral structures of all seven-coordinated Ln-dinuclear were categorized to be 7-MCO (monocapped-octahedron, point group: $C_{3v}$) using SHAPE software (Supplementary Note 3 and Supplementary Table 3). For the eight-coordinated Tb-polymer, the structure was categorized to be 8-SAP (Square antiprism, point group: $D_{4d}$) (Supplementary Table 4). The bidentate β-diketonate ligands display average Ln–O$_{tmh}$ distances of 2.294 and 2.332 Å in the dinuclear complexes and coordination polymers, respectively. The 4,4'-bpy ligands display average Ln–N distances of 2.585 and 2.619 Å in the dinuclear complexes and coordination polymers, respectively. The distances between the coordinated atoms and Ln(III) ions are larger in the eight-coordinate polymer than those in the seven-coordinate dinuclear complex. Larger distances result in a higher degree of freedom in the formation of metastable coordination polymers.

The 4,4'-bpy bridging ligand also exhibits key rotational and positional differences in the dinuclear and coordination polymer structures. The rotation angles of the py rings in 4,4'-bpy in the dinuclear complex and coordination polymer are 27° and 47°, respectively. The electronic states of the π-aromatic molecules depend strongly on their symmetrical structures[38]. The rotation angles and coordination distances in the structures of the 4,4'-bpy

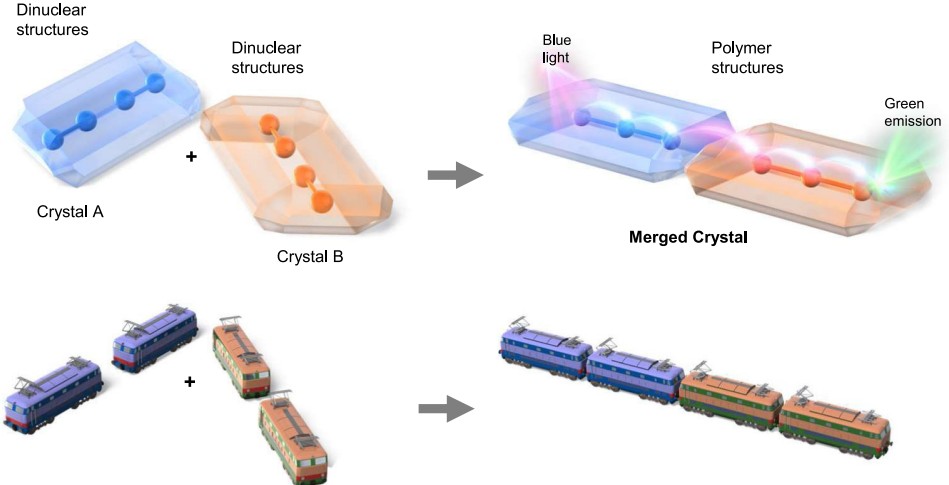

**Fig. 1 Conceptual design of this work.** Connection of two different emissive soft crystals, denoted molecular trains, for long-range energy migration.

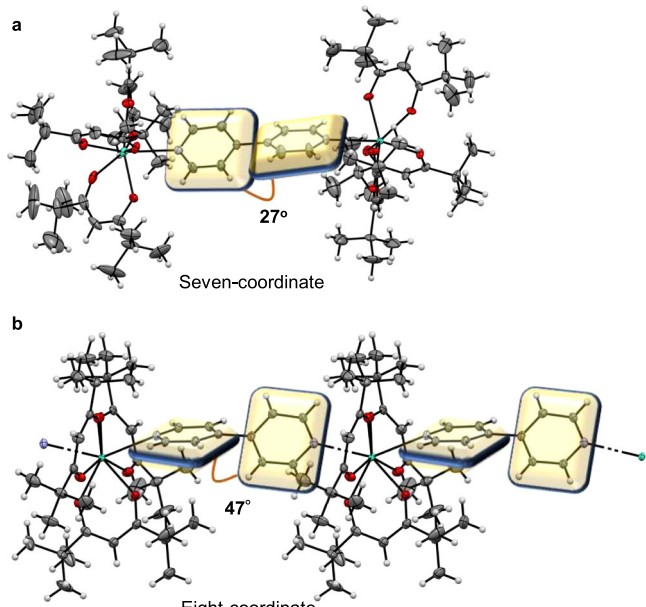

**Fig. 2 Structural analyses.** Views of **a** the [Tb₂(tmh)₆(4,4'-bpy)] dinuclear complex (Tb-dinuclear) and **b** the [Tb(tmh)₃(4,4'-bpy)]ₙ coordination polymer (Tb-polymer), showing 50% probability displacement ellipsoids. Gray spheres represent carbon; white spheres, hydrogen; red spheres, oxygen; blue spheres, nitrogen and green spheres, terbium. The planes of the bpy ligands are shown in yellow. tmh, 2,2,6,6-tetramethyl-3,5-heptanedionato; 4,4'-bpy, 4,4'-bipyridine.

and photosensitized tmh ligands may affect the energy transfer processes of the lanthanide complexes.

**Soft-crystal polymerization.** Single crystals of the dinuclear complexes and coordination polymers were successfully prepared. We then attempted in situ transformation from a dinuclear complex to a coordination polymer using an external stimulus. In a previous study, metastable eight-coordinate [Eu(tmh)₃(py)₂] was synthesized via py vaporization of the precursor [Eu₂(tmh)₆][29]. The Tb(III) coordination polymers also exhibited an eight-coordinate structure. To promote structural transformation from the dinuclear complex to the coordination polymer in the crystal media, py vaporization was performed (Fig. 3). The

Supplementary Video 1 shows the crystal transformation in detail.

The structural changes were analyzed using PXRD and XRD simulations based on the obtained single crystals. The transformed crystals were also subjected to infrared (IR) and elemental analyses. After the dinuclear structure was stored under py vapor, the PXRD pattern reveals significant changes (Fig. 3a). The new peaks are consistent with [Tb(tmh)₃(py)₂] (Supplementary Note 4 and Supplementary Fig. 4), suggesting that the transformed crystals include eight-coordinate [Tb(tmh)₃(py)₂] and 4,4'-bpy. The py exchange was investigated using density functional theory (DFT) calculations (B3LYP-D3, basis set: SDD (MWB28) for Tb(III) and 6–31 G(d) for other elements). Based on the optimized structure calculations, two pys are inserted in two steps (Fig. 3c, Supplementary Note 5, Supplementary Figs. 9, 10, and Supplementary Tables 5, 6). First, one py is inserted from the rear of 4,4'-bpy, followed by one py on top of 4,4'-bpy. The second insertion promotes the expansion of the coordination bonds between 4,4'-bpy and Tb(III) (2.603 Å), which are longer than those in [Tb₂(tmh)₆(4,4'-bpy)] (2.602 Å) and [Tb₂(tmh)₆(4,4'-bpy) + py] (2.589 Å) (Supplementary Figs. 10, 11 and Supplementary Table 6). The position of the second inserted py in conjunction with the expansion of coordination bonds suggests the ligand coordination exchange from 4,4'-bpy to py, forming of the intermediate [Tb(tmh)₃(py)₂].

After py removal under vacuum and drying, the PXRD peaks representing Tb-polymer (coordination polymer) and [Tb(tmh)₃(py)₁] (stoichiometric by-product) are observed (Fig. 3a and Supplementary Fig. 5). This transformation is not observed in methanol vapor, which was used in the synthesis (Supplementary Fig. 6). Thus, the formation of the eight-coordinate intermediate is critical in this transformation. The Tb(III)-N distances in the intermediate complex [Tb(tmh)₃(py)₂][39] are 2.603 Å, which is the mid-distance between the Tb-dinuclear Tb(III)–N distance of 2.585 Å and the Tb-polymer Tb(III)–N distance of 2.619 Å. The increasing Tb(III)–N distances and the formation of the eight-coordinate structure result in the necessary inter- and intramolecular rearrangements within the crystal. The reversibility of this transformation was confirmed under heating at 130 °C (Supplementary Note 6 and Supplementary Figs. 12–15). The same transformation behavior is observed for the Dy(III) dinuclear complex (Supplementary Figs. 7, 8), resulting in the formation of Dy(III) coordination polymers. This remarkable transformation leads to a unique in situ crystal-to-crystal surface connection at the molecular level.

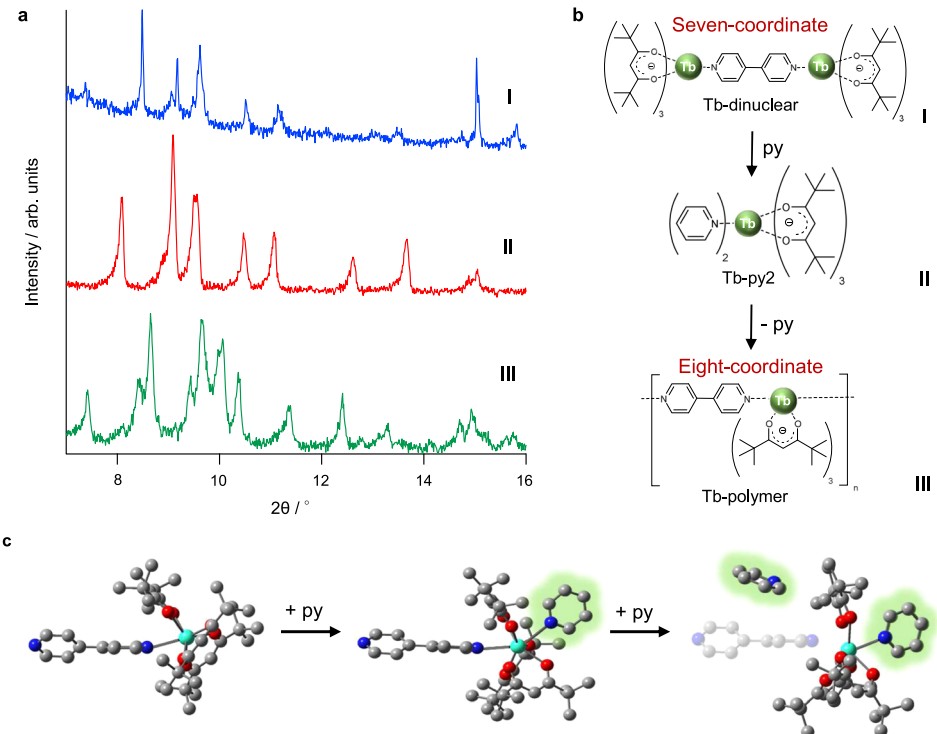

**Fig. 3 Structural polymerization. a** Powder X-ray diffraction patterns of the crystal sample at different steps. I: initial state (blue line: dinuclear structure); II: addition of pyridine (py) vapor (red line: conversion to [Tb(tmh)$_3$(py)$_2$] with uncoordinated 4,4'-bpy); III: elimination of py under vacuum (green line: transformation to coordination polymer). The third step includes the by-product [Tb(tmh)$_3$(py)$_1$][56]. **b** Illustration of the transformation. **c** Proposed structural transformation with py insertion. One Tb(tmh)$_3$ center of the dinuclear structure and hydrogen atoms are omitted for clarity. Gray spheres represent carbon; red spheres, oxygen; blue spheres, nitrogen and green spheres, terbium. Py molecules are highlighted in green. Tb, terbium; tmh, 2,2,6,6-tetramethyl-3,5-heptanedionato; 4,4'-bpy, 4,4'-bipyridine; py, pyridine.

**Energy migration in photonic molecular trains**. Based on the transformative polymerization, we attempted to merge independent monomer block crystals of the Tb(III) and Dy(III) dinuclear complexes. These two independent monomer crystals exhibit the same structures and transformabilities as the coordination polymer, resulting in block copolymerization on the crystal surface (Supplementary Video 1).

The energy transfer in the merged Tb(III) crystal-Dy(III) crystal was investigated by microspectroscopy for photophysical measurements. First, one Tb(III) dinuclear complex single crystal and one Dy(III) dinuclear complex single crystal were placed in contact on a quartz plate (Fig. 4a). After exposure (10 min) to and evaporation (30 min) of py vapor, the independent crystals were merged via polymerization. Their spatially resolved emission spectra on the crystal surface (positions 1, 2, 3, and 4 in Fig. 4a) were measured under Dy(III) direct excitation (Supplementary Note 7 and Supplementary Fig. 16: $\lambda_{ex} = 460 \pm 10$ nm). We also performed control experiments using separated crystals (Supplementary Figs. 17, 18), crystals connected at different surfaces (Supplementary Note 8 and Supplementary Fig. 19), grounded mixed crystals (Supplementary Note 10 and Supplementary Figs. 23, 24), and emission spectral measurements by a second harmonic generation ($\lambda = 452 \pm 2$ nm) of the mode-locked Ti:sapphire laser for the confirmation of the absence of directly excited Tb(III) emission in these conditions (Supplementary Note 9 and Supplementary Figs. 20–22).

Spatially resolved emissions at positions 1, 2, 3, and 4 of the unmerged Tb(III) crystal-Dy(III) crystal are shown in Fig. 4b. At positions 2, 3, and 4, a small Tb(III)-based emission at 550 nm is observed, with a strong Dy(III)-based emission at 570 nm (Fig. 4b). The spectra were normalized in their peak tops. The extended

pinhole area for detection (100 μm) in position 2 includes crystal edges of Dy(III) and Tb(III) compounds. The Tb(III) emission area in position 2 is smaller than in those in positions 3 and 4. We consider that the smaller Tb(III) emission band at position 2 is due to the smaller detection area and light scattering phenomena on the edge of the Tb(III) crystals. For the merged Tb(III) crystal-Dy(III) crystal after polymerization, we observe strong Tb(III) emissions at the center (position 4) of the Tb(III) crystal (Fig. 4c). The observed Tb(III) emission may only be explained by energy migration from the Dy(III) to Tb(III) crystals because only the Dy(III) ions are excited at $460 \pm 10$ nm ($^4I_{15/2} \leftarrow {}^6H_J$). Remarkably, the distance between the edge of the Dy(III) crystal and the center (position 4) of the Tb(III) crystal is ~150 μm. To the best of our knowledge, this energy migration distance is the longest reported for lanthanide coordination polymers or complex systems. This is consistent with the formation of polymeric luminescent crystal-to-crystal connections.

## Discussion

The energy migration in the merged crystal was investigated using photophysical measurements of each dinuclear complex and coordination polymer (excitation and emission spectra: Supplementary Note 11 and Supplementary Figs. 25 and 27). The emission lifetime and quantum yield of Tb-polymer ($\tau = 0.60$ ms, $\Phi = 64\%$) are approximately fourfold larger than those of Tb-dinuclear ($\tau = 0.15$ ms, $\Phi = 14\%$) (Table 1 and Supplementary Fig. 26). Previously, we reported that Tb(tmh)$_3$ centers with ancillary ligands show unique ligand-to-ligand charge transfer (LLCT) bands, which lead to emission quenching in Tb(III) complexes[33]. To investigate the energy transfer quenching in Tb-dinuclear and Tb-polymer, their temperature-dependent

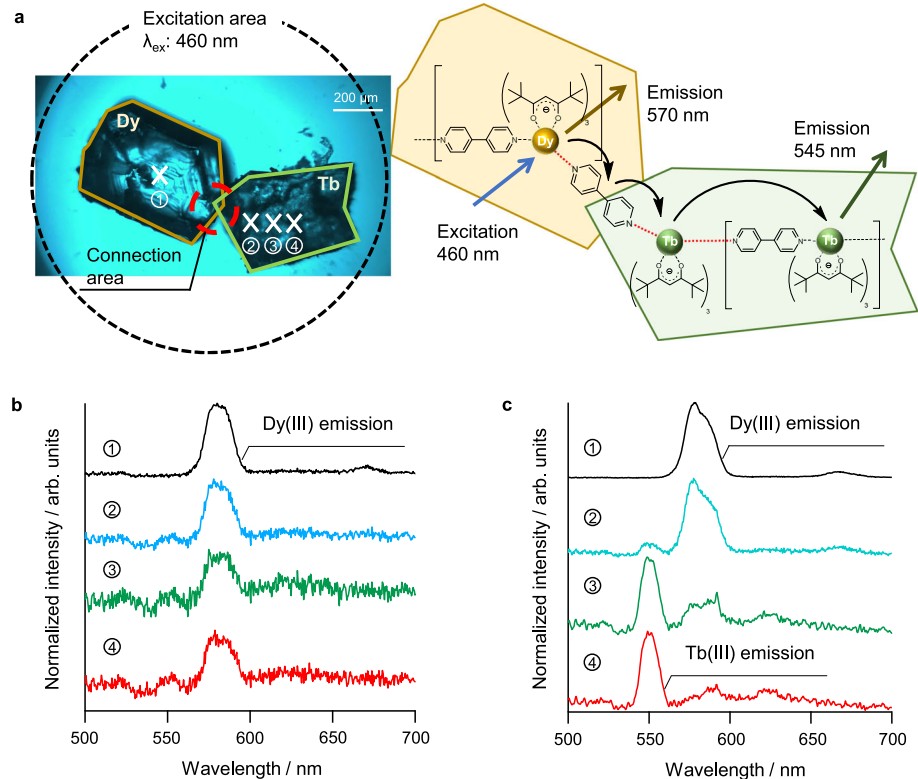

**Fig. 4 Energy migration in a merged Dy(III) crystal-Tb(III) crystal. a** Photophysical measurement schematic diagram. The crystal sample is excited using a 460 nm light source for direct Dy(III) excitation. Emission spectra were measured by spatially resolved microspectroscopy and normalized at their peak top. Position-dependent emission spectra in position 1 (black line), position 2 (blue line), position 3 (green line) and position 4 (red line) of the **b** dinuclear system (before transformation), and **c** coordination polymer (after transformation). Dy, dysprosium; Tb, terbium.

**Table 1 Photophysical properties of the dinuclear complexes and coordination polymers under ambient conditions.**

| | Formula | $\tau_{obs}$/ms | $\Phi_{ff}$ [a]/ % | $\Phi_{\pi\pi^\star}$ [b]/ % | $\eta_{sens}$ / % | $\Delta E$/cm$^{-1}$ | $A$/s$^{-1}$ |
|---|---|---|---|---|---|---|---|
| Tb-dinuclear | [Tb$_2$(tmh)$_6$ (4,4'-bpy)] | 0.15 ± 0.01 | 20 ± 5 | 14 ± 2 | 72 | 1.6 × 10$^3$ | 1.1 × 10$^7$ |
| Dy-dinuclear | [Dy$_2$(tmh)$_6$ (4,4'-bpy)] | 0.025 ± 0.001 | 6 ± 5 | 6 ± 2 | 99 | 3.5 × 10$^3$ | 3.2 × 10$^9$ |
| Tb-polymer | [Tb(tmh)$_3$ (4,4'-bpy)]$_n$ | 0.60 ± 0.02 | 64 ± 5 | 64 ± 2 | 99 | — | — |
| Dy-polymer | [Dy(tmh)$_3$ (4,4'-bpy)]$_n$ | 0.025 ± 0.001 | 7 ± 5 | 7 ± 2 | 99 | 2.9 × 10$^3$ | 2.5 × 10$^8$ |

[a]Measured using an integrating sphere (Tb(III) complexes: $\lambda_{ex}$ = 484 nm ($^5D_4 \leftarrow {}^7F_6$); Dy(III) complexes: $\lambda_{ex}$ = 455 nm ($^4I_{15/2} \leftarrow {}^6H_{15/2}$)). [b]Measured using an integrating sphere ($\lambda_{ex}$ = 360 nm). $\eta_{sens} = \Phi_{\pi\pi^\star} \times \Phi_{ff}^{-1} \times 100$. Arrhenius equation: $\ln (1/\tau_{obs} - 1/\tau_{200K}) = \ln (k_{ET}) = \ln (A) - (\Delta E/k_B T)$, where $k_B$ is the Boltzmann constant, $\Delta E$ is the activation energy, and $A$ is the frequency factor. Ambient temperature: 23 °C.

emission lifetimes were measured (Supplementary Fig. 29). Similarly, the Dy(III) complexes and coordination systems were also investigated (Supplementary Figs. 28 and 30).

The Tb-dinuclear emission lifetime shows a strong temperature dependence at 200–330 K, which is due to the energy transfer between the excited Tb(III) ion ($^5D_4$: 20,500 cm$^{-1}$)[40] and excited LLCT states[41] (23,190 cm$^{-1}$: time-dependent DFT (TD-DFT) calculations; Supplementary Note 12, Supplementary Fig. 31, and Supplementary Table 7) with activation energy ($\Delta E$) of 1.6 × 10$^3$ cm$^{-1}$. Conversely, the emission lifetime of Tb-polymer remains constant regardless of the temperature (100–390 K). The triplet energy levels of the tmh and 4,4'-bpy ligands of Tb-polymer are >25,000 cm$^{-1}$ [32], and the energy levels of the LLCT states are 24,240 cm$^{-1}$ based on TD-DFT calculation (Supplementary Fig. 32 and Supplementary Table 7). Therefore, Tb-polymer exhibits no close pathway for emission quenching.

For Dy-dinuclear, we also observe temperature-dependent emission quenching in the range 325–425 K because of their close energy levels (excited Dy(III) ion ($^4F_{9/2}$ = 21,100 cm$^{-1}$) and

LLCT: 23,190 cm$^{-1}$) (Supplementary Table 7) with a $\Delta E$ of 3.5 × 10$^3$ cm$^{-1}$. Dy-polymer also exhibits a decrease in the emission lifetime with increasing temperature from 325 to 390 K. These emission lifetime decreases are due to the energy transfer to the LLCT states (24,240 cm$^{-1}$), similar to the Tb(III) dinuclear complex system. At room temperature, however, no energy transfer is observed.

Based on these results, the proposed energy transfer processes of the merged and unmerged Tb(III) crystal-Dy(III) crystal systems are summarized in Fig. 5a, b. In the dinuclear system, energy transfer occurs either directly from the excited level of the Dy(III) ion ($^4F_{9/2}$ = 21,100 cm$^{-1}$) to the excited state of the Tb(III) ion ($^5D_4$: 20,500 cm$^{-1}$) via dipole-dipole interactions[42] or with the assistance of a close LLCT (23,190 cm$^{-1}$) state, although very inefficiently. However, even if populated, the Tb(III) excited states are easily quenched by the energy transfer to the LLCT state with a relatively small $\Delta E$ at room temperature (Table 1). In addition, energy transfer between Tb(III) ions is unfavorable, because the Tb(III)–Tb(III) distance (12 Å) is longer than the critical distance

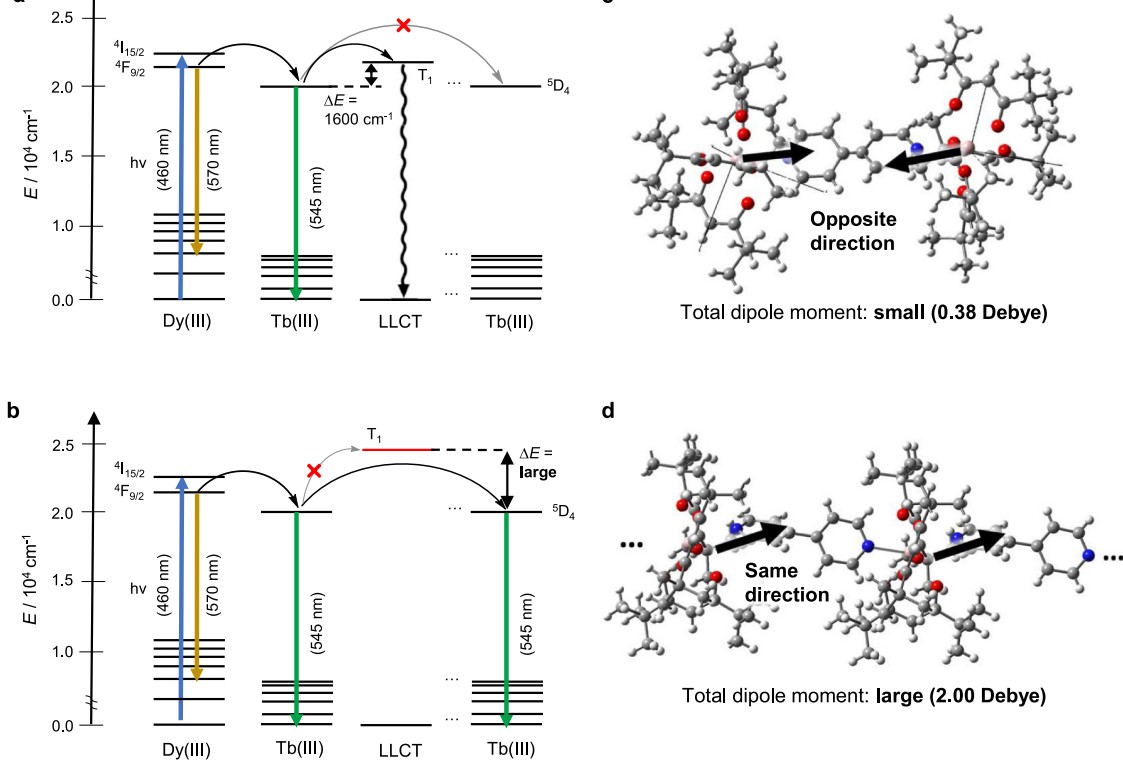

**Fig. 5 Energy migration process.** Energy diagram of the excited states in the **a** Dy(III)-Tb(III) dinuclear complex and **b** Dy(III)-Tb(III) coordination polymer based on the density functional theory calculations. The dipole moment calculated for each Tb(tmh)$_3$ center is based on the **c** dinuclear, and **d** polymer structures. Gray spheres represent carbon; white spheres, hydrogen; red spheres, oxygen; blue spheres, nitrogen and green spheres, terbium. Dy(III), dysprosium ion; Tb(III), terbium ion; tmh, 2,2,6,6-tetramethyl-3,5-heptanedionato; LLCT, ligand-to-ligand charge transfer.

(<10 Å) reported for Tb(III) excited-state energy transfer via dipole-dipole interactions in similar Tb(III) materials with small dipole moments (Fig. 5c)[43,44]. Therefore, the unmerged Tb(III) crystal-Dy(III) crystal displays very weak Tb(III)-based emission.

In the coordination polymer system, the LLCT excited states (24,240 cm$^{-1}$) are much higher than those in the dinuclear system, due to the changes in the rotation of the 4,4'-bpy and Ln–O$_{tmh}$ distances. This high energy level promotes efficient forward energy transfer from Dy(III) ions to Tb(III) ions without energy quenching. For the energy migration from Dy(III) ions to Tb(III) ions and between Tb(III) ions, this long-range energy transfer may be assisted by the delocalized excited states of the 4,4'-bpy and tmh ligands (Supplementary Note 13 and Supplementary Figs. 34, 35) via their strong excitonic and dipole interactions[45]. Thus, the excitonic interactions depend on the 1D connection provided by polymerization within the crystal, which is not observed in the dinuclear system (Fig. 5c, d, and Supplementary Figs. 33, 34). This behavior is compatible with strong Tb(III)–Tb(III) couplings[46], and therefore effective migration is observed in the coordination polymer system.

In summary, soft-crystal polymerization between Tb(III) and Dy(III) coordination centers on the crystal surfaces was achieved via transformation from a seven-coordinate dinuclear lanthanide complex to an eight-coordinate lanthanide coordination polymer. Single-crystal X-ray diffraction and PXRD confirmed the dynamic structural changes in the crystals, based on the change in coordination number of the lanthanide center. The dynamic polymerization of the lanthanide complexes in crystal media resulted in the merging of the Tb(III) and Dy(III) complexes due to their identical structures and transformations. After the connected crystals were formed via py vaporization, the system showed long-range energy transfer from the Dy(III) crystal

to the Tb(III) crystal (~150 μm). Quantum calculations and temperature-dependent emission lifetime measurements indicated ligand-assisted efficient energy transfer from the Dy(III) ions to Tb(III) in the coordination polymer systems. We also postulated that the formation of polymer chains enhanced the Tb(III)–Tb(III) couplings for long-range energy migrations.

Connected crystals prepared using two different crystals reveal new possibilities for the formation of novel crystal structures with new functionalities, such as crystal block copolymers and new printing techniques. Controlled mixing of different emissive crystals may also promote the formation of novel optical materials, such as nanocrystal p-n semiconductors, lasers, and fibers. Hetero-connected crystals exhibit the potential to broaden new areas of science and crystal engineering.

## Methods

**Materials.** All chemicals were of reagent grade and used without further purification. IR spectroscopy was performed using a JASCO FT/IR-4200 spectrometer, and elemental analyses were performed using a J-Science Lab JM 10 Micro Corder and an Exeter Analytical CE440 instrument. ESI-MS was performed using a JEOL JMS-T100LP instrument and a Thermo Scientific Exactive instrument.

**Synthesis of seven-coordinate dinuclear complex [Ln$_2$(tmh)$_6$(4,4'-bpy)] (Ln: Tb(III) or Dy(III)).** The precursor complex, [Ln$_2$(tmh)$_6$], was synthesized according to a literature method[47]. [Ln$_2$(tmh)$_6$] (Ln: Tb(III) or Dy(III); 0.21 g, 0.15 mmol) and 4,4'-bpy (15 mg, 0.10 mmol) were dissolved in methanol at a molar ratio of 3:2. The mixture was stirred for 24 h at room temperature. The solvent was removed via rotary evaporation. The residue was recrystallized from methanol, yielding transparent single crystals.

[Tb$_2$(tmh)$_6$(4,4'-bpy)]: yield: 0.190 g (92 %). IR (ATR) $\tilde{v} = 2836$–3002 (m, C–H), 1589 (m, C–N), 1569 cm$^{-1}$ (s, C = O); elemental analysis (%): calcd. for C$_{76}$H$_{122}$N$_2$O$_{12}$Tb$_2$: C, 58.01; H, 7.81; N, 1.78; found: C, 57.38; H, 7.95; N, 1.32.

[Dy$_2$(tmh)$_6$(4,4'-bpy)]: yield: 0.150 g (71 %). IR (ATR) $\tilde{v} = 2836$–3002 (m, C–H), 1589 (m, C–N), 1569 cm$^{-1}$ (s, C = O); elemental analysis (%): calcd. for

$C_{76}H_{122}N_2O_{12}Dy_2 + 3\ CH_3OH$: C, 56.58; H, 8.05; N, 1.67; found: C, 56.78; H, 7.77; N, 1.51.

**Synthesis of eight-coordinate coordination polymer [Tb(tmh)$_3$(4,4′-bpy)]$_n$.** [Tb$_2$(tmh)$_6$] (0.21 g, 0.15 mmol) and 4,4′-bpy (90 mg, 0.60 mmol) were dissolved in methanol at a mass ratio of 1:2. The solution was stirred for 4 h at room temperature, and the solvent was removed via rotary evaporation. The residue was recrystallized from methanol, yielding transparent single crystals. Dy(III) coordination polymers were not obtained using this method.

Yield: 0.188 g (89 %). IR (ATR) $\tilde{v}$ = 2836–3002 (m, C–H), 1587 (m, C–N), 1574 cm$^{-1}$ (s, C = O). ESI-MS (m/z): calcd. for $C_{32}H_{46}N_2O_4T$ b [M − tmh]$^+$: 681.27, found: 681.27; elemental analysis (%): calcd. for $C_{43}H_{65}N_2O_6Tb$: C, 59.71; H, 7.58; N, 3.24; found: C, 59.35; H, 7.61; N, 3.18.

**Structural transformation (soft-crystal polymerization).** The seven-coordinate dinuclear complex was polymerized in crystal form via exposure to py vapor for 10 min, followed by drying under air for 1 h. The vaporization was performed in a Petri dish containing a saturated py atmosphere. The obtained crystals were evaluated using PXRD, IR, and elemental analyses.

[Tb(tmh)$_3$(4,4′-bpy)]$_n$*: IR (ATR) $\tilde{v}$ = 2836–3002 (m, C–H), 1589 (m, C–N), 1571 cm$^{-1}$ (s, C = O); elemental analysis (%): calcd. for $C_{43}H_{65}N_2O_6Tb_1$ (Tb-polymer) + $C_{114}H_{186}N_3O_{18}Tb_3$ (3 Tb-py1): C, 58.41; H, 7.84; N, 2.17; found: C, 58.02; H, 7.84; N, 2.10.

[Dy(tmh)$_3$(4,4′-bpy)]$_n$*: IR (ATR) $\tilde{v}$ = 2836–3002 (m, C–H), 1591 (m, C–N), 1570 cm$^{-1}$ (s, C = O); elemental analysis (%): calcd. for $C_{43}H_{65}N_2O_6Dy_1$ (Dy-polymer) + $C_{114}H_{186}N_3O_{18}Dy_3$ (3 Dy-py1): C, 58.15; H, 7.80; N, 2.16; found: C, 57.80; H, 7.68; N, 2.19.

**Crystal-to-crystal connection.** One single-crystal block (~500 μm) of a seven-coordinate dinuclear Tb(III) complex and one single-crystal block (~500 μm) of a seven-coordinate dinuclear Dy(III) complex were placed on top of a thin 5 × 5 mm quartz glass with their respective smallest sides in contact. They were exposed to py vapor for 10 min, and then to dry air for 1 h for structural polymerization. The obtained crystal system was merged to form a soft-crystal block copolymer.

**Optical measurements.** Detailed photophysical measurement descriptions for the excitation and emission spectra, emission lifetimes (temperature-dependent), and emission quantum yields are available in the Supplementary Information. The energy migration of the Tb(III)-Dy(III) connected crystal was measured using a spatially resolved microspectroscopic system for emission detection at specific regions of the merged crystal. The spatial resolution is estimated to be 20 μm because we employed the objective lens 5× and a pinhole with a diameter of 100 μm on the focal image position before the spectrometer. Expanded light detection is observed up to a radius of 100 μm on the focal image position, probably due to crystalline optical disorder. A halogen light spectrally filtered to the wavelength of 460 ± 10 nm using a band-pass filter was employed as an excitation source. The exposure time of the spectrometer was set to 1 s and the excitation light was cut by a long-pass edge filter at 510 nm. All data are available in the Figshare database.

**Crystallography.** The X-ray crystal structures and crystallographic data of [Ln$_2$(tmh)$_6$(4,4′-bpy)] (Ln: Tb(III) and Dy(III)) and [Ln(tmh)$_3$(4,4′-bpy)]$_n$ (Ln: Tb(III)) are shown in Fig. 2 and Supplementary Table 1. Detailed measurement information is available in the Supplementary Information.

**DFT calculations.** Quantum chemical calculations were performed with the *Gaussian 16* package[48]. Geometry optimization was carried out using DFT with B3LYP-D3 functional[49–53]. Grimme's D3 dispersion correction[53] is incorporated since the intramolecular dispersion interaction plays a significant role in the stabilization of these systems including many aromatic rings. The 6–31 G(d) basis set was used for all elements, except for Tb(III), in which SDD (MWB28 for complexes including pyridine, MWB54 for Tb-dinuclear, and Tb-polymer) basis set was used[54,55]. At the calculation of ground state (S$_0$) for Tb-dinuclear and Tb-polymer, the positions of hydrogens were optimized and others are fixed to the crystal structure. The lowest triplet state (T$_1$) of them were investigated using their partially-optimized S$_0$ structures and by TD-DFT calculation with the same functional and basis set. All data are available in the Figshare database.

## Data availability

The single-crystal data generated in this study have been deposited in The Cambridge Crystallographic Data Center database [https://www.ccdc.cam.ac.uk/structures/]. The PXRD (https://doi.org/10.6084/m9.figshare.19514353.v1), DFT (https://doi.org/10.6084/m9.figshare.19514449.v1), steady-state (https://doi.org/10.6084/m9.figshare.19514380.v1; https://doi.org/10.6084/m9.figshare.19514377.v1; https://doi.org/10.6084/m9.figshare.19514359.v1) and spatially resolved microspectroscopic (https://doi.org/10.6084/m9.

figshare.19514386.v1) photophysical raw data are available in the Figshare database [https://figshare.com/browse].

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

## Acknowledgements
This work was supported by the Japan Society for the Promotion of Science (JSPS) KAKENHI Grant Numbers JP18H04504 (H.U.), JP20H04661 (H.U.), JP18H02041 (Y.H.), JP20H04653 (Y.H.), JP20H05197 (Y.H.), JP20K21201 (Y.H.), JP19H04556 (Y.K.), JP20H02748 (Y.K.), JP17H06373 (H.G.), and JP21K05105 (N.N.) of the Ministry of Education, Culture, Sports, Science, and Technology (MEXT) of Japan. This work was also supported by the Institute for Chemical Reaction Design and Discovery (ICReDD), established by the World Premier International Research Center Initiative (WPI) and JSPS Research Fellow Number 19J20713 (P.P.F.R.) of MEXT, Japan. The authors express sincere thanks to professor H. Ito and Dr. J. Mingoo of Hokkaido University and Dr. T. Seki of Shizuoka University for single-crystal X-ray diffraction. We also thank Mr. Takuma Nakai from Hokkaido University for his assistance in the video production.

## Author contributions
P.P.F.R. performed all the syntheses, crystal-to-crystal connection, and transformations, steady-state photophysical measurements, X-ray analyses and wrote the paper under the supervision of S.S., Y.K., K.F., and Y.H. H.O. and H.U. were responsible for the PXRD assignment of transformed molecules. N.N. and H.G. performed the DFT calculations, and K.I. and K.U. supported the spatially resolved microspectroscopic measurements. All authors reviewed the manuscript and approved the final version.

## Competing interests
The authors declare no competing interest.
