## [Peer Review File · Nature Communications]

Photonic molecular trains in soft-crystal polymerization of transformative lanthanide coordination centresREVIEWER COMMENTS

Reviewer #1 (Remarks to the Author):

Photochromic soft crystals have been studied for many years, and many significant works have been reported (Chem. Soc. Rev. 2004, 33, 579–588; Chem. Rev. 2014, 114, 12174–12277). This work synthesized two crystals [Ln₂(tmh)₆(4,4'-bpy)] and [Ln(tmh)₃(4,4'-bpy)]_n, and their structure can be transformed under py vapor or vacuum. They also found the connected Tb(III) crystal-Dy(III) crystal showed photon energy transfer from the Dy(III) centre to the Tb(III) centre under blue light excitation (selective Dy(III) centre excitation: 460 ± 10 nm). This work may be interesting, but there still need improvement and more explorations should be carried out to prove their conclusion. Hence, I think this work is difficult to meet the criteria of nature communications with the current version.

Comments are listed as follows:

1. Two crystals were placed in contact on a quartz plate. Whether the distance between two crystals is close enough to form a chemical band or not? Even if the bond is formed, the authors need more in-situ characterization to prove it. Furthermore, the characterizations are based on a single crystal transformation, and the transition between two different metal-based crystals requires more explorations.
2. Why the strong Tb(III) emissions at the Tb(III) crystal (position 2, Fig. 4c) is not observed? which has the shortest energy migration distance from the Dy(III) to Tb(III) crystals. The author should give a reasonable explanation.
3. In order to enhance the efficiency of energy migration, more contact areas need to be studied, such as line or surface contact, rather than just point contact. In addition, if the Dy(III) to Tb(III) crystals are grinded into powder and mixed, and further exposure (10 min) to and evaporation (30 min) of py vapour, will this phenomenon be more noticeably observed in fluorescence detector (not spatially resolved microspectroscopy)?

Reviewer #2 (Remarks to the Author):

The authors have introduced a novel technique of polymerization by simple chemistry. At the moment, the experiment and theory are not commensurate for publication in Nature Comm and two suggestions are made to increase the scientific content of the manuscript.

1. The excitation of the sample using 460±10 nm is not accurate. I suggest to use the 457 nm or 465 nm lines of an argon-ion laser – naturally checking for the absence of Tb³⁺ emission from the Tb³⁺ sample. The energy transfer can then be more certainly assigned.
2. It is not adequate to use Al³⁺ in the DFT calculations. The authors are requested to use Tb³⁺ with the relevant effective core potential.

If these guidelines are followed, the manuscript may be suitable for publication.

Some minor points -

For the non-specialist, the introduction gives some complicated chemical names (which are not properly cited – for example 2,2,6,6-Tetramethyl-3,5-heptanedionato – but the chemical formulae are actually simple. Please give the chemical formulae of the ligands employed either in the manuscript or SI. The Figure 1 is good since it clearly shows the aim of the work.

SI

Please give reflux temperature.

Please show the structure of Ln₂(tmh)₆.

Fig 17 write excitation for excited.

Fig 27 – I am concerned about the purity of chemicals employed in this study (stated as reagent grade). There are large bands due to other lanthanide ions in the spectra of Gd³⁺ and similar impurities present for Tb and Dy compounds would affect the conclusions drawn.

I do not make any statements about the crystallographic data.

Reviewer #3 (Remarks to the Author):

General comment:

The manuscript by da Rosa, Kitagawa, Shoji, Oyama, Imaeda, Nakayama, Fushimi, Uekusa, Ueno, Goto and Hasegawa present the synthesis, structure, optical characterizations and time-dependent DFT calculations for a soft-crystal copolymerization with Dy(III) and Tb(III) lanthanides. This study is in adequacy with the "Nature Communications" journal. I recommend publication after minor revisions; some improvements could be realized before publication. I would like to point out that this is a very nice piece of work that is a great opening to the design of new emissive compounds by soft crystal way.

Comments and Minor points:

1- In the supporting information; for lifetime figure, "us" could be replaced by "µs". There are a small mistake for the attribution form Tb(III) transition, at 545 nm, this assignment is 5D4 → 7D5 and not 7F3 (figure S17, 18 and 21). In the table S1 and S2, uncertainties on crystallographic parameters could be added.

2- The lifetime measurement has it been measured several times? The values could be indicated with uncertainty errors in the tables.

3- For the quantum yields, which is the reproducibility? What is the uncertainty? In view of the uncertainty of about 10% that exists for quantum yield measurements (integrating sphere), the values the values could be indicated with an error bar.

4- In the dinuclear complexes and in the coordination polymer what is the geometry around the metal centers Ln? This geometry could be simply identify by SHAPE software.

5- In the cif and especially checkcif files, alerts B and C should be justified.

Review letter

Reviewer #1

Photochromic soft crystals have been studied for many years, and many significant works have been reported (Chem. Soc. Rev. 2004, 33, 579–588; Chem. Rev. 2014, 114, 12174–12277). This work synthesized two crystals [Ln₂(tmh)₆(4,4'-bpy)] and [Ln(tmh)₃(4,4'-bpy)]_n, and their structure can be transformed under py vapor or vacuum. They also found the connected Tb(III) crystal-Dy(III) crystal showed photon energy transfer from the Dy(III) centre to the Tb(III) centre under blue light excitation (selective Dy(III) centre excitation: 460 ± 10 nm). This work may be interesting, but there still need improvement and more explorations should be carried out to prove their conclusion. Hence, I think this work is difficult to meet the criteria of nature communications with the current version.

<Ans.>

We thank reviewer 1 for kind comments and suggestions. We revised our manuscript with novel experiments based on your suggestions.

Comments are listed as follows:

1. Two crystals were placed in contact on a quartz plate. Whether the distance between two crystals is close enough to form a chemical band or not? Even if the bond is formed, the authors need more in-situ characterization to prove it. Furthermore, the characterizations are based on a single crystal transformation, and the transition between two different metal-based crystals requires more explorations.

<Ans.>

We thank reviewer 1 for the suggestions and comments. A Supplementary Video 1 has been included to show the chemical bond formed between the crystals following the transformation. Two crystals are placed in contact before transformation, however once transformation is complete, the coordination bond formation connects the two crystals and they become one. They're even strong enough to be lifted without breaking.

In order to understand the molecules and bonds rearrangements, we attempted a Molecular Dynamics(MD) calculation of the transformation, but the number of molecules and atoms exponentially increased the complexity of the system, and the calculations did not converge enough to obtain a desirable result. We were able to use geometry optimization calculations to predict the formation of pyridine bonds in the dinuclear structure at the first stage of the transformation. Please check <Supp. Info., pages S14-S16 (Intermediate formation in py vaporisation)>

2. Why the strong Tb(III) emissions at the Tb(III) crystal (position 2, Fig. 4c) is not observed? which has the shortest energy migration distance from the Dy(III) to Tb(III) crystals. The author should give a reasonable explanation.

<Ans.>

Thank you for sharing your concern. The spectra were normalized to their peak tops in position 2, 3 and 4 of Fig. 4c. The extended pin-hole area for detection (100 μm) in position 2 includes crystal edges of Dy(III) and Tb(III) compounds. This pin-hole area detects strong Dy(III) emission and weak Tb(III) emission. The Tb(III) emission area in position 2 is smaller than in those in position 3 and 4. We consider that smaller Tb(III) emission band at position 2 is due to the smaller detection area and light scattering phenomena on the edge of the Tb(III) crystals. We included this information in the main manuscript.

<page 11, line 204>

After: "At positions 2, 3, and 4, a small Tb(III)-based emission at 550 nm is observed, with strong Dy(III)-based emission at 570 nm (**Error! Reference source not found.**)"

We added: "The spectra were normalized in their peak tops. The extended pin-hole area for detection (100 μm) in position 2 includes crystal edges of Dy(III) and Tb(III) compounds. The Tb(III) emission area in position 2 is smaller than in those in position 3 and 4. We consider that smaller Tb(III) emission band at position 2 is due to the smaller detection area and light scattering phenomena on the edge of the Tb(III) crystals."

<page 12, Fig. 4 caption>

After: "Emission spectra were measured by spatially resolved microspectroscopy"

We added: "and normalized at their peak top."

3. In order to enhance the efficiency of energy migration, more contact areas need to be studied, such as line or surface contact, rather than just point contact.

<Ans.>

We would like to thank reviewer 1 for sharing your suggestions. In the manuscript, we connected the crystals via the point contact. According to the reviewer's suggestion, we performed face-to-face surface contact experiments, additionally. The results reveal that, more than the sort of contact, the face on which the crystals are bonded has a direct impact on the efficiency of energy migration. The Tb(III)-based emission intensity on the crystal connection at (-1 2 1) face (parallel connection to the direction of 4,4'-bpy) is stronger than that at (1 1 0) face (perpendicular connection to the direction of 4,4'-bpy). We propose that structural transformation following the 4,4'-bpy ligand in the crystal connection enhances the linkage between the crystals, and facilitates the energy transfer process. These results and their explanations were added in the Supplementary Information and mentioned in the main manuscript.

<page 11, line 197>

After “We also performed control experiments using separated crystals (Supplementary Figs. 15-16)...”
We added: “, crystals connected at different surfaces (Supplementary Fig. 17), grounded mixed crystals (Supplementary Figs. 21-22) and emission spectral measurements by a second harmonic generation ($\lambda = 452 \pm 2$ nm) of the mode-locked Ti:sapphire laser for the confirmation of absence of directly excited Tb(III) emission in these conditions (Supplementary Figs. 18-20: $\lambda_{\text{ex}} = 452 \pm 2$ nm).”

<Supp. Info., page S24>

Energy migration experiments (different surface connections)

The energy migration of the Tb(III)-Dy(III) connected crystal was measured using different surface connections to investigate the dependence in the energy migration efficiency. Point connected crystals at (-1 2 1) face showed stronger Tb(III) emission than point connected crystals at (1 1 0) face in two positions (Supplementary Fig. 17c-d). Here, (-1 2 1) face follows the same direction as 4,4'-bpy (Supplementary Fig. 17e). On the other hand, (1 1 0) face is almost perpendicular to the direction of 4,4'-bpy in the crystal (Supplementary Fig. 17f). The structural transformation depends on the direction of 4,4'-bpy since it is the coordinated ligand connecting Ln(III) ions. We propose that structural transformation following the 4,4'-bpy ligand in the crystal connection enhances the linkage between the crystals and facilitates the energy transfer process.

Supplementary Fig. 17. a) Photophysical measurement schematic diagram of point connected crystals. b) Photophysical measurement schematic diagram of surface connected crystals. The crystal samples are excited using a 460 nm light source for direct Dy(III) excitation. Emission spectra were measured by spatially resolved microspectroscopy at the cross indication. Connected faces are represented by arrows with their correspondent labels. Emission spectra of transformed crystals normalized at 570 nm (Dy(III) based emission) in c) position 3 (closer to Dy(III) crystal) and d) position 4 (middle of Tb(III) crystal). e) Perspective views of $[\text{Tb}_2(\text{tmh})_6(4,4'\text{-bpy})]$ dinuclear complex showing (-1 2 1) plane in red. f) Perspective views of $[\text{Tb}_2(\text{tmh})_6(4,4'\text{-bpy})]$ dinuclear complex showing (1 1 0) plane in red.

3. In addition, if the Dy(III) to Tb(III) crystals are grinded into powder and mixed, and further exposure (10 min) to and evaporation (30 min) of py vapour, will this phenomenon be more noticeably observed in fluorescence detector (not spatially resolved microspectroscopy)?

<Ans.>

According to the reviewer's request, we carried out the additional experiments with mixed Tb(III) and Dy(III) powders. After transformation, we were able to observe energy transfer from directly excited Dy(III) crystals to Tb(III) crystals. Due to the larger surface contact between the crystals, the energy transfer phenomenon was slightly more noticeable than in a connected crystal system. This finding was noted in the main manuscript and added to the Supplementary Information.

<page 11, line 197>

After "We also performed control experiments using separated crystals (Supplementary Figs. 15-16)..." we added: ", crystals connected at different surfaces (Supplementary Fig. 17), grounded mixed crystals (Supplementary Figs. 21-22) and emission spectral measurements by a second harmonic generation ($\lambda = 452 \pm 2$ nm) of the mode-locked Ti:sapphire laser for the confirmation of absence of directly excited Tb(III) emission in these conditions (Supplementary Figs. 18-20: $\lambda_{\text{ex}} = 452 \pm 2$ nm)."

<Supp. Info., page S29>

Energy migration experiments (mixed powder)

The energy migration was also investigated in powder conditions. Here, the Dy(III) and Tb(III) crystals are grinded into powder and mixed, and further transformed using pyridine vaporization. The energy migration was evaluated using excitation (Supplementary Fig. 22) and emission spectra (Supplementary Fig. 21) recorded on a HORIBA Fluorolog-3 spectrofluorometer and corrected for the response of the detector system. A xenon light diffracted using a diffraction grating with 0.5 nm slit to the wavelength of 452 ± 2 nm coupled with a 450 nm long-pass filter was employed as an excitation source for the emission spectra. Emission detection was set at 541 nm with 0.6 nm slit coupled with a 540 ± 5 nm bandpass filter for the excitation spectra.

Supplementary Fig. 21. Emission spectra of the Dy-polymer powder (after transformation from Dy-dinuclear: black line) and Tb-polymer/Dy-polymer mixed powder (after transformation from Tb-dinuclear and Dy-dinuclear mixed powders: red line) excited at 452 ± 2 nm. The spectra were normalized at their peak top.

The Tb(III)-based emission at around 545 nm was observed at 452 ± 2 nm (Dy(III) direct excitation) excitation along with the Dy(III)-based emission in the mixed system. The Tb(III)-based emission cannot be observed in the 100% Dy-polymer powder. This experiment indicates that energy migration between Dy(III) and Tb(III) powder occurs effectively. In order to further confirm the energy migration, we measured the excitation spectra based on the Tb(III) emission (545 nm).

Supplementary Fig. 22. Excitation spectra of the Tb-polymer powder (after transformation from Tb-dinuclear: black line) and Tb-polymer/Dy-polymer mixed powder (after transformation from Tb-dinuclear and Dy-dinuclear mixed powders: red line) detected at 541 ± 2 nm. The spectra were normalized at 484 nm (Tb(III) direct excitation: $^5D_4 \leftarrow ^7F_6$; peak not shown).

A novel excitation band at around 453 nm is observed in the mixed system compared to the sample with 100% Tb-polymer powder. This band is assigned to be $^4I_{15/2} \leftarrow ^6H_{15/2}$ of Dy(III) ion. This excitation band strongly supports that Tb(III) emission was enabled by Dy(III) powder energy migration.

Reviewer #2

The authors have introduced a novel technique of polymerization by simple chemistry. At the moment, the experiment and theory are not commensurate for publication in Nature Comm and two suggestions are made to increase the scientific content of the manuscript.

<Ans.>

We thank reviewer 2 for kind comments and suggestions. We revised our manuscript with novel experiments based on your suggestions.

(1) *The excitation of the sample using 460±10 nm is not accurate. I suggest to use the 457 nm or 465 nm lines of an argon-ion laser naturally checking for the absence of Tb³⁺ emission from the Tb³⁺ sample. The energy transfer can then be more certainly assigned.*

<Ans.>

We would like to thank reviewer 2 for the suggestion We used a second harmonic generation of the mode-locked Ti:sapphire laser as an excitation light source to conduct the energy transfer experiment. We were unable to conduct the experiment using an argon laser due to a lack of it, but the second harmonic generation of the mode-locked Ti:sapphire laser proved to be effective in the selective excitation of Dy(III) crystals. In our confocal microscope system, the excitation and detection sources were separated by at least 10 μm in order to investigate the energy migration between crystals. This distance between excitation and detection positions could not be further separated due to confocal limitation. Tb(III) emission was only detected when the Dy(III) and Tb(III) crystals were connected. These results are introduced in the Supporting Information, while the original experiment is remained in the main manuscript for its clear detection of Tb(III) emission.

<page 11, line 197>

After “We also performed control experiments using separated crystals (Supplementary Figs. 15-16)...” we added: “, crystals connected at different surfaces (Supplementary Fig. 17), grounded mixed crystals (Supplementary Figs. 21-22) and emission spectral measurements by a second harmonic generation ($\lambda = 452 \pm 2$ nm) of the mode-locked Ti:sapphire laser for the confirmation of absence of directly excited Tb(III) emission in these conditions (Supplementary Figs. 18-20: $\lambda_{\text{ex}} = 452 \pm 2$ nm).”

<Supp. Info., page S26>

Energy migration experiments (laser excitation)

The energy migration of the Tb(III)-Dy(III) connected crystal was measured using a spatially resolved microspectroscopic system for emission detection at specific regions of the merged crystal. The spatial resolution is estimated to be 20 μm because we employed the objective lens 5× and a pinhole with a diameter of 100 μm on the focal image position before the spectrometer. However, an expanded light detection is observed up to a radius of 100 μm on the focal image position, probably due to crystalline optical disorder Here, a second harmonic generation (SHG; 452 ± 2 nm) of the mode-locked Ti:sapphire laser (904 nm, 80 MHz, 100 fs) was employed as an excitation source (Supplementary Figs. 18-19). The exposure time of the spectrometer was set to 10 s and the excitation light was cut by a long-pass edge filter at 480 nm. The excitation and detection sources were separated by at least 10 μm in order to investigate the energy migration between crystals (Supplementary Fig. 20).

Supplementary Fig. 18. Excitation spectra of **Dy-polymer** (dark yellow line) detected at 571 nm and **Tb-polymer** (dark green line) detected at 541 nm in comparison to the profile of the excitation light (SHG of the mode-locked Ti:sapphire laser: $\lambda_{\text{ex}} = 452 \pm 2$ nm).

Supplementary Fig. 19. Emission of **Tb-dinuclear** excited at 452 nm (SHG of the mode-locked Ti:sapphire laser: $\lambda_{\text{ex}} = 452 \pm 2$ nm).

Supplementary Figs. 18 and 19 confirm the absence of absorption and emission of Tb(III) crystals under the laser excitation (SHG of the mode-locked Ti:sapphire laser: $\lambda_{\text{ex}} = 452 \pm 2$ nm).

Supplementary Fig. 20. a) Photophysical measurement schematic diagram for laser excited experiment. The crystal sample is excited using a 452 ± 2 nm laser light source for direct Dy(III) excitation. Emission spectra were measured by spatially resolved microspectroscopy. b) Emission spectra of the dinuclear system (before transformation: black line) and coordination polymer (after transformation: red line) at the specified position.

The unmerged Tb(III) crystal-Dy(III) crystal (before transformation) showed only Dy(III)-based emission at 570 nm (Supplementary Fig. 20b). For the merged Tb(III) crystal-Dy(III) crystal, we observe Tb(III) emissions along with Dy(III) emissions (Supplementary Fig. 20b), derived from the large radius of the spatially resolved microspectroscopy. The observed Tb(III) emission may only be explained by energy migration from the Dy(III) to Tb(III) crystals because only the Dy(III) ions are excited at 452 ± 2 nm using a laser system. This result is in line with results obtained in the main manuscript.

(2) *It is not adequate to use Al3+ in the DFT calculations. The authors are requested to use Tb3+ with the relevant effective core potential. If these guidelines are followed, the manuscript may be suitable for publication.*

<Ans.>

I thank reviewer 2 for the important indication. We recalculated using Tb3+ with SDD (MWB54) basis set. The energy levels and properties were not significantly different from the prior findings. As a result, the discussion remained unchanged. The recalculated values and additional calculation methods are added in the main manuscript and Supporting Information, as follows:

<page 20, line 377>

Old "Excited states were investigated using TD-DFT based on the B3LYP functional with the 6-31G(d) basis set. Al(III) ions were used instead of Tb(III) ions to reduce the computational cost.³⁴"

New: "Geometry optimization was carried out using DFT with B3LYP-D3 functional.⁵¹⁻⁵⁵ The Grimme's D3 dispersion correction⁵⁵ is incorporated since the intramolecular dispersion interaction plays significant role for the stabilization of these systems including many aromatic rings. The 6-31G(d) basis set was used for all elements, except for Tb(III), in which SDD (MWB28 for complexes including pyridine, MWB54 for Tb-dinuclear and Tb-polymer) basis set was used.^{56,57}

At the calculation of ground state (S_0) for Tb-dinuclear and Tb-polymer, the positions of hydrogens were optimized and others are fixed to the crystal structure. The lowest triplet state (T_1) of them were investigated using their partially-optimized S_0 structures and by TD-DFT calculation with the same functional and basis set."

51. Becke, A. D. Density-functional exchange-energy approximation with correct asymptotic behavior. *Phys. Rev. A* **38**, 3098-3100 (1988).

52. Lee, C., Yang, W., Parr, R. G. Development of the Colle-Salvetti correlation-energy formula into a functional of the electron density. *Phys. Rev. B*, **37**, 1988, 785-789 (1988).

53. Becke, A. D. Density-functional thermochemistry . III . The role of exact exchange. *Chem. Phys.* **98**, 5648-5652 (1993).

54. Stephens, P. J., Devlin, F. J., Chabalowski, C. F. & Frisch, M. J. Ab Initio Calculation of Vibrational Absorption and Circular Dichroism Spectra Using Density Functional Force Fields. *J. Phys. Chem.* **98**, 11623-11627 (1994).

55. Grimme, S., Antony, J., Ehrlich, S. & Krieg, H. A consistent and accurate ab initio parametrization of density functional dispersion correction (DFT-D) for the 94 elements H-Pu. *Chem. Phys.* **132**, 154104 (2010).

56. Dolg, M., Stoll, H. & Preuss, H. A combination of quasirelativistic pseudopotential and ligand field calculations for lanthanoid compounds. *Theor. Chim. Acta* **85**, 441-450 (1993).

57. Dolg, M., Stoll, H., Savin, A. & Preuss, H. Energy-adjusted pseudopotentials for the rare earth elements. *Theor. Chim. Acta* **75**, 173-194 (1989).

<lines 247, 251, 256, 259, 265, 279, Figure 5>

Corrected the energy levels in text and figure to the new results.

<Supp. Info., Supplementary Figs. 29-30>

Erased the old molecular orbital representations and added the new representations based on the new result using Tb³⁺.

<Supp. Info., Supplementary Table 7>

Supplementary Table 7. Summary of calculated T_1 states of **Tb-dinuclear** and **Tb-polymer**.

Complex	T_1 / nm	T_1 / cm^{-1}	Character	$\Delta E (T_1-^5D_4) / \text{cm}^{-1}$	$\Delta E (T_1-^4F_{9/2}) / \text{cm}^{-1}$
Tb-dinuclear	431.2	23,190	Strong LLCT	2,690	2,090
Tb-polymer	412.6	24,240	Weak LLCT	3,740	3,140

Tb(III) excited state (5D_4 : 20,500 cm^{-1}); Dy(III) excited state ($^4F_{9/2}$: 21,100 cm^{-1}).⁷

Some minor points –

For the non-specialist, the introduction gives some complicated chemical names (which are not properly cited – for example 2,2,6,6-Tetramethyl-3,5-heptanedionato – but the chemical formulae are actually simple. Please give the chemical formulae of the ligands employed either in the manuscript or SI. The Figure 1 is good since it clearly shows the aim of the work.

<Ans.>

Thank you for the nice comment and suggestions. I corrected the chemical names in the main manuscript and added the chemical formula of the ligands in the Supporting Information.

<page 2, line 22; page 3, line 63; Fig. 2; Fig. 3; Fig. 5; Supplementary Figs. 1, 2, 3, 7, 8, 9, 13, 31, 32, 33>

Corrected the chemical names for 2,2,6,6-Tetramethyl-3,5-heptanedionato.

<Supp. Info., page S2>

before: Ln(III): Tb(III) or Dy(III), tmh: tetramethylheptandionato, 4,4'-bpy: 4,4'-bipyridine.

after: Ln(III): Tb(III) or Dy(III), tmh: 2,2,6,6-tetramethyl-3,5-heptandionato ($\text{C}_{11}\text{H}_{19}\text{O}_2$), 4,4'-bpy: 4,4'-bipyridine ($\text{C}_{10}\text{H}_8\text{N}_2$).

SI

Please give reflux temperature.

Please show the structure of $\text{Ln}_2(\text{tmh})_6$.

Fig 17 write excitation for excited.

<Ans.>

Thank you for the suggestions. I added and corrected the following information as requested.

<Supp. Info., page S2>

Conditions: All reactions were performed under reflux at 60 °C.

Structure of $\text{Ln}_2(\text{tmh})_6$ (CCDC 755629):⁴

1. Stabnikov, P. A., Zharkova, G. I., Smolentsev, A. I., Pervukhina, N. v. & Krisyuk, V. v. Structure and properties of terbium(III) dipivaloylmethanate and its adducts with Bipy and Phen. *Journal of Structural Chemistry* **52**, 560–567 (2011).

<Supp. Info., Supplementary Figs. 23 and 25 captions>

Supplementary Fig. 23. Excitation (left) and emission (right) spectra of **Tb-dinuclear** (blue line) and **Tb-polymer** (brown line) in solid state at room temperature. Excitation spectra were recorded with emission at 545 nm ($^5\text{D}_4 \rightarrow ^7\text{F}_5$) and normalized at 484 nm (Tb(III) direct excitation: $^5\text{D}_4 \leftarrow ^7\text{F}_6$; peak not shown). Emission spectra were excited at 350 nm and normalized at their peak tops.

Supplementary Fig. 25. Excitation (left) and emission (right) spectra of **Dy-dinuclear** (dark yellow line) and **Dy-polymer** (red line) in solid state at room temperature. Excitation spectra were recorded with emission at 571 nm ($^4\text{F}_{9/2} \rightarrow ^6\text{H}_{13/2}$). Emission spectra were excited at 350 nm. They were normalized at their peak top.

Fig 27 – I am concerned about the purity of chemicals employed in this study (stated as reagent grade). There are large bands due to other lanthanide ions in the spectra of Gd^{3+} and similar impurities present for Tb and Dy compounds would affect the conclusions drawn. I do not make any statements about the crystallographic data.

<Ans.>

Thank you for pointing out your concern. We also express the same concern regarding the purity of Ln(III) ions used. In the case of Gd(III) complexes, the starting $\text{GdCl}_3 \cdot 6\text{H}_2\text{O}$ reactant showed 99.9% purity. Due to

comparable physical features, Gd(III) ions are easily contaminated with Eu(III) and Tb(III) ions. Furthermore, these coordination compounds have exceptional photosensitization efficiencies (near 100%) for Tb(III) ions, allowing them to be emitted even at 0.1 percent concentration. The emission derived from β -diketonate and pyridine ligands in the Gd(III) complexes is also relatively weak, which compares to the Tb(III) emission. We included this comment in the Supplementary Information.

For Tb(III) and Dy(III) complexes and coordination polymers, we also used starting reactants with 99.9% purity. Since we did not observe any absorption or emission bands derived from other Ln(III) ions at 100-420 K range, we consider that the impurities are not of great concern for this system. Even with a presence of 0.1% of other Ln(III) ions, we strongly believe that those other ions' excited states are barely involved in the total energy transfer process.

<Supp. Info., page S43>

After: "Therefore, the emission for Gd(III) based complexes is derived from the singlet and triplet states of the ligands."

We added: "In the case of Gd(III) complexes, the starting $\text{GdCl}_3 \cdot 6\text{H}_2\text{O}$ reactant showed 99.9% purity. Due to comparable physical features, Gd(III) ions are easily contaminated with Eu(III) and Tb(III) ions. Furthermore, these coordination compounds have exceptional photosensitization efficiencies (near 100%) for Tb(III) ions, allowing them to be emitted even at 0.1% concentration. The emission derived from β -diketonate and pyridine ligands in the Gd(III) complexes is also relatively weak, which compares to the Tb(III) emission."

Reviewer #3

The manuscript by da Rosa, Kitagawa, Shoji, Oyama, Imaeda, Nakayama, Fushimi, Uekusa, Ueno, Goto and Hasegawa present the synthesis, structure, optical characterizations and time-dependent DFT calculations for a soft-crystal copolymerization with Dy(III) and Tb(III) lanthanides. This study is in adequacy with the "Nature Communications" journal. I recommend publication after minor revisions; some improvements could be realized before publication. I would like to point out that this is a very nice piece of work that is a great opening to the design of new emissive compounds by soft crystal way.

<Ans.>

Thank you for your kind comments and suggestions.

(1) In the supporting information; for lifetime figure, “us” could be replaced by “μs”. There are a small mistake for the attribution form Tb(III) transition, at 545 nm, this assignment is 5D4 → 7D5 and not 7F3 (figure S17, 18 and 21). In the table S1 and S2, uncertainties on crystallographic parameters could be added.

<Ans.>

Thank you for the thorough observations. The corrections were made accordingly. In Supplementary Tables 1-2, the uncertainties on crystallographic parameters were added.

<Supp. Info., Supplementary Fig. 26>

<Supp. Info., Supplementary Fig.23 caption>

“...Excitation spectra were recorded with emission at 545 nm ($^5D_4 \rightarrow ^7F_5$) and normalized...”

<Supp. Info., Supplementary Fig. 24 caption>

“Detection wavelength was set to 545 nm ($^5D_4 \rightarrow ^7F_5$).”

<Supp. Info., Supplementary Fig. 27 caption>

“a) Emission lifetime temperature-dependent derived from $^5D_4 \rightarrow ^7F_5$ for **Tb-dinuclear...**”

<Supp. Info., Supplementary Table 1>

Supplementary Table 1. Crystallographic data

	[Dy ₂ (tmh) ₆ (4,4'-bpy)]	[Tb ₂ (tmh) ₆ (4,4'-bpy)]	[Tb(tmh) ₃ (4,4'-bpy)] _n
Chemical formula	C ₇₆ H ₁₂₂ N ₂ O ₁₂ Dy ₂	C ₇₆ H ₁₂₂ N ₂ O ₁₂ Tb ₂	C ₄₃ H ₆₅ N ₂ O ₆ Tb
F000	1640.0	3272.0	864.89
Crystal system	Triclinic	Triclinic	Monoclinic
Space group	P -1 (no. 2)	P -1 (no. 2)	P 2 ₁ / c (no. 14)
a / Å	9.74380(10)	11.9730(2)	11.8300(3)
b / Å	11.9357(2)	19.4913(3)	34.6307(7)
c / Å	37.0588(7)	36.9619(5)	12.0379(3)
α / deg	96.3150(10)	93.4770(10)	90
β / deg	93.6220(10)	96.3170(10)	118.160(3)
γ / deg	101.3280(10)	101.407(2)	90
Volume / Å ³	4184.37(12)	8373.9(2)	4347.9(2)
Z	2	2	4
T / °C	-150	-150	-150
R ₁ ^[a] / %	4.99	8.57	3.31
wR ₂ ^[b] / %	13.92	18.04	9.98

[a] $R_1 = \Sigma ||F_0| - |F_c|| / \Sigma |F_0|$.

[b] $wR_2 = [\Sigma w(F_0^2 - F_c^2)^2 / \Sigma w(F_0^2)^2]^{1/2}$.

<Supp. Info., Supplementary Table 2>

Supplementary Table 2. Crystallographic data for dinuclear polymorphs

	[Dy ₂ (tmh) ₆ (4,4'-bpy)] - A	[Tb ₂ (tmh) ₆ (4,4'-bpy)] - A	[Dy ₂ (tmh) ₆ (4,4'-bpy)] - B	[Tb ₂ (tmh) ₆ (4,4'-bpy)] - B
Chemical formula	C ₇₆ H ₁₂₂ N ₂ O ₁₂ Dy ₂	C ₇₆ H ₁₂₂ N ₂ O ₁₂ Tb ₂	C ₇₆ H ₁₂₂ N ₂ O ₁₂ Dy ₂	C ₇₆ H ₁₂₂ N ₂ O ₁₂ Tb ₂
F000	1640.0	1636.0	820.0	818.0
Crystal system	Monoclinic	Monoclinic	Triclinic	Triclinic
Space group	P 2 ₁ / n (no. 14)	P 2 ₁ / n (no. 14)	P -1 (no. 2)	P -1 (no. 2)
a / Å	9.9349(2)	9.9131(4)	9.7319(2)	9.7401(2)
b / Å	21.1409(5)	21.1864(7)	11.9099(3)	11.9471(2)
c / Å	19.7546(4)	19.8214(7)	18.6753(5)	18.6244(5)
α / deg	90	90	89.770(2)	89.515(2)
β / deg	101.748(2)	101.838(4)	80.651(2)	80.412(2)
γ / deg	90	90	79.025(2)	78.8630(10)
Volume / Å ³	4062.20(15)	4074.4(3)	2095.95(9)	2096.10(8)
Z	4	4	2	2
T / °C	-150	-150	-150	-150
R ₁ ^[a] / %	2.47	8.14	4.31	6.85
wR ₂ ^[b] / %	5.82	16.32	11.05	18.88

[a] $R_1 = \Sigma ||F_0| - |F_c|| / \Sigma |F_0|$.

[b] $wR_2 = [\Sigma w(F_0^2 - F_c^2)^2 / \Sigma w(F_0^2)^2]^{1/2}$.

(2) *The lifetime measurement has it been measured several times? The values could be indicated with uncertainty errors in the tables.*

(3) *For the quantum yields, which is the reproducibility? What is the uncertainty? In view of the uncertainty of about 10% that exists for quantum yield measurements (integrating sphere), the values the values could be indicated with an error bar.]*

<Ans.>

Indeed the lifetime measurements were measured several times and the quantum yields show uncertainties related to the measurement. For lanthanide direct excitation, in which the light absorption is small, higher uncertainties are obtained, around 5% after 5 measurements. For the photosensitized

excitation, the uncertainties are relatively small, around 2% after 5 measurements. The uncertainty errors were added in the tables accordingly. Thank you for your suggestion.

<page 13, Table 1>

Table 1. Photophysical properties of the dinuclear complexes and coordination polymers under ambient conditions.

	Formula	$\tau_{\text{obs}} / \text{ms}$	$\Phi_{\text{ff}}^{\text{a}} / \%$	$\Phi_{\pi\pi^*}^{\text{b}} / \%$	$\eta_{\text{sens}} / \%$	$\Delta E / \text{cm}^{-1}$	A / s^{-1}
Tb-dinuclear	[Tb ₂ (tmh) ₆ (4,4'-bpy)]	0.15 ± 0.01	20 ± 5	14 ± 2	72	1.6 × 10 ³	1.1 × 10 ⁷
Dy-dinuclear	[Dy ₂ (tmh) ₆ (4,4'-bpy)]	0.025 ± 0.001	6 ± 5	6 ± 2	99	3.5 × 10 ³	3.2 × 10 ⁹
Tb-polymer	[Tb(tmh) ₃ (4,4'-bpy)] _n	0.60 ± 0.02	64 ± 5	64 ± 2	99	—	—
Dy-polymer	[Dy(tmh) ₃ (4,4'-bpy)] _n	0.025 ± 0.001	7 ± 5	7 ± 2	99	2.9 × 10 ³	2.5 × 10 ⁸

a: Measured using an integrating sphere (Tb(III) complexes: $\lambda_{\text{ex}} = 484 \text{ nm}$ (⁵D₄ ← ⁷F₆); Dy(III) complexes: $\lambda_{\text{ex}} = 455 \text{ nm}$ (⁴I_{15/2} ← ⁶H_{15/2})). b: Measured using an integrating sphere ($\lambda_{\text{ex}} = 360 \text{ nm}$). $\eta_{\text{sens}} = \Phi_{\pi\pi^*} \times \Phi_{\text{ff}}^{-1} \times 100$. Arrhenius equation: $\ln(1/\tau_{\text{obs}} - 1/\tau_{200\text{K}}) = \ln(k_{\text{ET}}) = \ln(A) - (\Delta E/k_{\text{B}}T)$, where k_{B} is the Boltzmann constant, ΔE is the activation energy, and A is the frequency factor. Ambient temperature: 23 °C.

(4) *In the dinuclear complexes and in the coordination polymer what is the geometry around the metal centers Ln? This geometry could be simply identify by SHAPE software.*

<Ans.>

Thank you for your suggestion. We performed SHAPE calculations and found the closest geometry around the Ln(III) centres. The pseudo-coordination polyhedral structures of all seven-coordinated **Ln-dinuclear** were categorized to be 7-MCO (monocapped-octahedron, point group: C_{3v}). For the eight-coordinated **Tb-polymer**, the structure was categorized to be 8-SAP (Square antiprism, point group: D_{4d}). We added this result in the main manuscript with more detail explanation in the Supporting Information.

<page 7, line 118>

Before: “The bidentate β-diketonate ligands ...” we added: “The pseudo-coordination polyhedral structures of all seven-coordinated **Ln-dinuclear** were categorized to be 7-MCO (monocapped-octahedron, point group: C_{3v}) using SHAPE software (Supplementary Table 3). For the eight-coordinated **Tb-polymer**, the structure was categorized to be 8-SAP (Square antiprism, point group: D_{4d}) (Supplementary Table 4).”

SHAPE calculations

In order to obtain the closest coordination geometry around the Ln(III) centres, continuous shape measurements (CShM) calculations were performed using SHAPE.⁴ The S_{CShM} criterion represents the degree of deviation from ideal coordination structure, and is given by the following equation:

$$S_{\text{CShM}} = \min \frac{\sum_k^N |Q_k - P_k|^2}{\sum_k^N |Q_k - Q_0|^2} \times 100$$

where Q_k is the vertices of an actual structure, Q_0 is the center of mass of an actual structure, P_k is the vertices of an ideal structure, and N is the number of vertices. The estimated S_{CShM} values of the Ln(III) complexes are summarized in Supplementary Table 3 and Supplementary Table 4. Based on the minimum value of S_{CShM} , the pseudo-coordination polyhedral structures of all seven-coordinated dinuclear Ln(III) complexes were categorized to be 7-MCO (monocapped-octahedron, point group: C_{3v}). For the eight-coordinated Tb(III) coordination polymer, the structure was categorized to be 8-SAP (Square antiprism, point group: D_{4d}). The coordination geometry of the seven-coordinated dinuclear complexes is more asymmetric than that of eight-coordinated Tb(III) coordination polymer.

Supplementary Table 3. Calculated S_{CShM} values for seven-coordinated Ln(III) complexes

Complex	$S_{7\text{-MCO}}$	$S_{7\text{-MCTP}}$	$S_{7\text{-PBP}}$
Tb-dinuclear1	0.63128	1.31333	11.17713
Tb-dinuclear2	0.83202	0.94406	9.66405
Dy-dinuclear1	0.49079	1.53628	11.18781
Dy-dinuclear2	0.79385	0.81259	10.28508
Tb-dinuclearA	0.96587	1.02542	9.82093
Dy-dinuclearA	0.91235	1.01750	9.84488
Tb-dinuclearB	0.57754	1.25175	10.91711
Dy-dinuclearB	0.53983	1.32892	10.92708

S_{7-MCO} , S_{7-MCTP} , S_{7-PBP} are S values calculated for monocapped-octahedron (Point group: C_{3v}), monocapped-trigonal prism (Point group: C_{2v}) and pentagonal-bipyramid (Point group: D_{5h}), respectively.

Supplementary Table 4. Calculated S_{CSHM} values for Tb(III) coordination polymer

Complex	S_{8-SAP}	S_{8-TDD}	S_{8-CU}
Tb-polymer	0.68469	2.84109	10.47320

S_{8-SAP} , S_{8-TDD} , S_{8-CU} are S values calculated for square antiprism (Point group: D_{4d}), triangular dodecahedron (Point group: D_{2d}) and cube (Point group: O_h), respectively.

4. a: Casanova, D., Cirera, J., Llundell, M., Alemany, P., Avnir, D. & Alvarez, S. Minimal Distortion Pathways in Polyhedral Rearrangements, *Journal of the American Chemical Society*, **126**, 1755-1763 (2004). b: Pinsky, M. & Avnir, D. Continuous Symmetry Measures. 5. The Classical Polyhedra. *Inorganic Chemistry* **37**, 5575–5582 (1998).

(5) *In the cif and especially checkcif files, alerts B and C should be justified.*

<Ans.>

Thank you for your suggestion. We revised the structure analyses and added the respective justifications for each alert B and C. Please check the new cif files.

REVIEWER COMMENTS

Reviewer #1 (Remarks to the Author):

The authors have addressed all my previous comments properly, and I would recommend its publication.

Reviewer #2 (Remarks to the Author):

The authors have responded to the numerous points raised by the Reviewers and have updated and changed the manuscript accordingly. I believe that it is now acceptable for publication in Nature Comm. There is a typo for "separate" in the video.

Reviewer #3 (Remarks to the Author):

The revisited manuscript by da Rosa, Kitagawa, Shoji, Oyama, Imaeda, Nakayama, Fushimi, Uekusa, Ueno, Goto and Hasegawa present the synthesis, structure, optical characterizations and time-dependent DFT calculations for a soft-crystal copolymerization with Dy(III) and Tb(III) lanthanides. The authors have taken the time to answer all the questions of the referees. Moreover, they also performed additional manipulations sometimes on their own initiative to meet the expectations of the referees. For all these reasons, the manuscript should now be accepted in Nature Communications.

Review letter

Reviewer #1

The authors have addressed all my previous comments properly, and I would recommend its publication.

<Ans.>

We thank reviewer 1 for recommending the publication. We appreciate your previous suggestions since it improved the quality of this paper.

Reviewer #2

The authors have responded to the numerous points raised by the Reviewers and have updated and changed the manuscript accordingly. I believe that it is now acceptable for publication in Nature Comm. There is a typo for "separate" in the video.

<Ans.>

We thank reviewer 2 for recommending the publication. We appreciate your previous suggestions since it improved the quality of this paper. Thank you for pointing out the typo in the video. It was corrected.

Reviewer #3

The revisited manuscript by da Rosa, Kitagawa, Shoji, Oyama, Imaeda, Nakayama, Fushimi, Uekusa, Ueno, Goto and Hasegawa present the synthesis, structure, optical characterizations and time-dependent DFT calculations for a soft-crystal copolymerization with Dy(III) and Tb(III) lanthanides. The authors have taken the time to answer all the questions of the referees. Moreover, they also performed additional manipulations sometimes on their own initiative to meet the expectations of the referees. For all these reasons, the manuscript should now be accepted in Nature Communications.

<Ans.>

We thank reviewer 3 for recommending the publication. We appreciate your previous suggestions since it improved the quality of this paper.